# Simulation of the critical nitrogen dilution curve in Jiangxi double-cropped rice region based on leaf dry matter weight

Chun Ye[1,2], Ying Liu[1]*, Jizhong Liu[1]*, Yanda Li[2‡], Binfeng Sun[2‡], Shifu Shu[2‡], Luofa Wu[2‡]

**1** School of Mechanical and Electrical Engineering, Nanchang University, Nanchang, China, **2** Institute of Agricultural Engineering/Jiangxi Province Engineering Research Center of Intelligent Agricultural Machinery Equipment/Jiangxi Province Engineering Research Center of Information Technology in Agriculture, Jiangxi Academy of Agricultural Sciences, Nanchang, China

☯ These authors contributed equally to this work.
‡ YL, BS, SS and LW also contributed equally to this work.
* lying@ncu.edu.cn (YL); jizhongl@163.com (JL)

## Abstract

In order to investigate the feasibility of using rice critical nitrogen concentration as a nitrogen nutrition diagnosis index, a two-year positioning field gradient experiment using four rice varieties and four nitrogen levels (0, 75, 150, 225 kg·ha$^{-1}$ for early rice; 0, 90, 180, 270 kg·ha$^{-1}$ for late rice) was conducted for early and late rice. The critical dilution curves (N$_c$%) of the double-cropped rice based on leaf dry matter (LDM) were constructed and verified using the field data. Two critical nitrogen dilution curves and nitrogen nutrition indexes (NNI) of rice LDM were constructed for early rice [N$_c$% = 2.66LDM$^{-0.79}$, $R^2$ = 0.88, NNI ranged between 0.29–1.74, and the average normalized root mean square error ($n$-RMSE = 19.35%)] and late rice [N$_c$% = 7.46LDM$^{-1.42}$, $R^2$ = 0.91, NNI was between 0.55–1.53, and the average ($n$-RMSE = 15.14%)]. The relationship between NNI and relative yield was a quadratic polynomial equation and suggested that the optimum nitrogen application rate for early rice was sightly smaller than 150 kg·ha$^{-1}$, and that for late rice was about 180 kg·ha$^{-1}$. The developed critical nitrogen concentration dilution curves, based on leaf dry matter, were able to diagnose nitrogen nutrition in the double-cropped rice region.

## Introduction

As one of the three major food crops in the world, rice has seen a 3.3-fold increase in yield in the past 50 years, mainly as a result of major resource investment into production, including fertilizer, water, pesticide, and advanced crop breeding technology [1]. Nitrogen is the main nutrient element for crop growth and development. It plays an important role in the synthesis of proteins, nucleic acids, phospholipids, and other organic nitrogen compounds that are necessary for plant growth, improving crop photosynthesis, and enhancing assimilation products [2]. In the double-cropped rice planting environment of Jiangxi, China, which is mainly large

**Data Availability Statement:** All relevant data are within the manuscript and its Supporting information files.

**Funding:** This research was jointly supported by the Key-Area Research and Development Program of Jiangxi Province [20202BBFL63046, 20202BBFL63044], The National Natural Science Foundation of China (41961048), and the National Key Research and Development Program of China (2016YFD0300608).

**Competing interests:** The authors have declared that no competing interests exist. We thank LetPub (www.letpub.com) for its linguistic assistance during the preparation of this manuscript.

in scale and small in area, excessive chemical fertilizer application is common and has led to increasingly prominent environmental pollution. For instance, the water, soil, and the atmosphere are all seriously polluted by nitrogen fertilizer [3]. Therefore, it is important that the application of nitrogen fertilizer is optimized in the double-cropped rice area to improve the utilization rate of nitrogen fertilizer, protect the environment, and realize sustainable agricultural development.

For the rapid and accurate evaluation of crop nitrogen status, most recent studies have used chlorophyll meters and spectral remote sensing image technology for monitoring and diagnosis [4–6]. However, these methods generally suffer from unreliable diagnostic results when nitrogen is absorbed as a substance [7]. To solve the above problems, Ulrich [8] proposed the concept of a critical nitrogen concentration; that is, the minimum nitrogen concentration required for the maximum growth of crops. Greenwood, Lemaire et al. [9–11] then proposed new models for $C_3$ and $C_4$ crops that used the nitrogen nutrition index (NNI) to diagnose the nitrogen concentration in grain and forage in order to optimize fertilizer application. However, the model was based on the average value of multiple experiments. When the model was applied to a new climate region, some differences in crop variety parameters were observed. It is thus necessary that localized parameters are studied. The critical nitrogen concentration dilution curve was established based on plant dry matter and plant nitrogen concentration and is used to determine the lowest nitrogen concentration when the aboveground dry matter reaches the maximum growth rate. At present, the model has been applied to potato [12], tomato [13], rape [14], corn [15], wheat [16], and other crops. The critical nitrogen concentration dilution curve model usually combines two individual and population characteristics of plant nitrogen concentration and aboveground biomass. In recent years, agronomists have proposed some new nitrogen concentration models based on the combination of plant nitrogen concentration or image canopy coverage [17], leaf area index [18], or population characteristics of different parts of the plant [19], which have been proved to be good quantitative indicators of crop growth nitrogen nutrition status.

In this study, the critical nitrogen dilution curve of double-cropped rice in the Jiangxi region was established based on leaf dry matter mass to diagnose the nitrogen nutrition status of main rice varieties under different nitrogen application rates. The critical nitrogen concentration curve model was also used to analyze leaves from different genotypes of high-quality rice and determine the critical nitrogen concentration. The purpose of this study was to explore a more convenient and efficient method for nitrogen monitoring in double-cropped rice and provide a theoretical basis for the development of portable diagnostic instruments and field robots.

## Materials and methods

### Experimental design

**Experiment 1.** The experiment was carried out from 2019 to 2020 in Gao'an city, Jiangxi province, China. The experimental station was located at 28°25'27" N and 115°12'15" E. The cultivated layer of the experimental field contained 38.80 g·kg$^{-1}$ organic matter, 2.53 g·kg$^{-1}$ total nitrogen, 42.4 mg·kg$^{-1}$ ammonium nitrogen, 1.04 mg·kg$^{-1}$ nitrate nitrogen, 16.78 mg·kg$^{-1}$ available phosphorus, and 120.1 mg·kg$^{-1}$ available potassium and had a pH of 5.5.

The early rice varieties were 'Zhongjiazao 17' and 'Changliangyou 173', were the main varieties in Jiangxi province, which were treated with four nitrogen levels (0, 75, 150, and 225 kg·ha$^{-1}$, represented by N0, N75, N250, and N225, respectively) three times, resulting in a total of 24 treatments. The ratio of base fertilizer to tiller fertilizer to panicle fertilizer was 5:3:2. Phosphate fertilizer (60 kg·ha$^{-1}$) was applied as base fertilizer once, and potassium fertilizer

**Table 1. Basic information on the two experiments.**

| Data point classification | Classification of rice | Varieties | Transplanting /harvesting date | Nitrogen fertilizer levels (kg·ha$^{-1}$) |
|---|---|---|---|---|
| **Experiment 1 in 2019** | Early rice | 'Zhongjiazao 17', 'Chanliangyou 173' | 4.23/7.14 | Early rice: 0, 75, 150, 225 |
| | Late rice | 'Fumeizhan', 'Taiyouhang 1573' | 7.28/10.30 | Late rice: 0, 90, 180, 270 |
| **Experiment 1 in 2020** | Early rice | 'Zhongjiazao 17', 'Changiangyou 173' | 4.19/7.11 | |
| | Late rice | 'Fumeizhan', 'Taiyouhang 1573' | 7.21/10.24 | |
| **Experiment 2 in 2019** | Early rice | 'Zaoxian 618', 'Ganxin 203' | 4.27/7.10 | |
| | Late rice | 'Taiyou 398', 'Wufengyou T025' | 7.21/10.28 | |

(120 kg·ha$^{-1}$) was applied together with the same proportion of nitrogen fertilizer. The plot covered an area of 30 m$^2$ (5 m × 6 m). Three plants were transplanted in a north–south direction. The plots were separated by ridges, covered with plastic film, and irrigated independently.

'Fumeizhan' and 'Taiyouhang 1573' were used as the late rice varieties, which all belong to a large area of cultivated varieties in Jiangxi. A two-factor randomized block design was used in the experiment. Four nitrogen levels (0, 90, 180, and N270 kg·ha$^{-1}$, represented by N0, N90, N180, and N270, respectively) were set. Nitrogen fertilizer was applied three times, including 50% base fertilizer, 30% tillering fertilizer, and 20% panicle fertilizer. The row spacing between the two plants was 14 cm × 24 cm. Other cultivation measures were consistent with local high-yield cultivation practices.

**Experiment 2.** The experiment was carried out in 2019 in Jie'bu town, Xingan County, Yi'chun city, China. The experimental site was located at 115°21'3.87" E and 27°45'17.65" N. The cultivated layer of the experimental field contained 28.0 g·kg$^{-1}$ organic matter, 127.0 mg·kg$^{-1}$ alkali hydrolyzable nitrogen, 29.0 mg·kg$^{-1}$ available phosphorus, and 121.0 mg·kg$^{-1}$ available potassium and had a pH of 5.5. The early rice varieties were 'Zaoxian618' and 'Ganxin 203', and the late rice varieties were 'Taiyou 398' and 'Wufengyou T025'. The test details and design are shown in Table 1.

## Plant sampling and determination of N content in the tissues

**Plant sampling.** Samples were drawn from five separate hills, and sampling commenced from 16 or 18 days after transplanting in 2019 and 2020. Samples were taken from the tillering stage, jointing stage, booting stage, heading stage, and full heading stage of rice. Five representative rice plants were collected in each plot. Once back at the laboratory, the stems and leaves were separated. The dry matter of the leaves per hectare (t·ha$^{-1}$) was calculated based on the leaf dry matter per plant multiplied by the number of tillers per hectare. The rice leaves, stems, and panicles were dried at 105°C for 30 min and then dried to a constant weight at 80°C. Then the rice leaves, stems, and panicles were crushed using a small pulverizer, and the nitrogen content of the rice was determined by a Kjeldahl nitrogen analyzer. Finally, the plant nitrogen concentration (PNC) was obtained by multiplying the dry matter mass of leaf, stem, and panicle with the corresponding nitrogen concentration and summing, then dividing by the total dry matter mass of the rice plant.

**Determination of yield.** At harvest time, the yield was calculated based on three repetitions of a single plot (1 m$^2$) and a single harvest. The average yield of the three plots was used to calculate the treatment yield and was then converted into yield per hectare (t·ha$^{-1}$). The relative yield (RY) was the ratio of each processing yield divided by the highest yield of the quarter.

## Model construction and validation

**Model construction.**   We followed the concept of the $N_c$ dilution curve developed by Justes [20], and the modeling steps were as follows: (*a*) analysis of variance (ANOVA) of the leaf day matter of each sampling and its corresponding nitrogen concentration, which were divided into a nitrogen-limited group and non-nitrogen limited group; (*b*) linear fitting of leaf dry matter (LDM) and PNC data of the nitrogen-limited treatment; (*c*) a vertical line was used to represent the average LDM between the non-nitrogen limited treatment, which was the maximum LDM of this sampling; (*d*) the intersection coordinate between the oblique line and vertical line of each sampling date was used to determine the nitrogen concentration value. The dilution curve equation of critical nitrogen effect based on LDM is as follows:

$$N_c\% = aLDM^{-b}, \tag{1}$$

where *a* represents the critical nitrogen concentration when the dry matter of the leaves is 1, and *b* represents the slope of the critical nitrogen concentration dilution curve.

**Model calibration.**   The data from Experiment 2 were used to test the model. The root mean square error (RMSE) and relative root mean square error (n-RMSE) were used to evaluate the reliability of the model, and the 1:1 relationship between the observed value and the simulated value was drawn to show the fitting degree and prediction effect of the model.

$$\text{RMSE} = \sqrt{\frac{1}{n} \times \sum_{i=1}^{n} (P_i - O_i)^2} \tag{2}$$

$$n - \text{RMSE} = \text{RMSE}/\overline{O_i} \times 100\% \tag{3}$$

In the equations, *n* is the number of samples, $P_i$ is the predicted value, $\overline{Pi}$ is the average value of the predicted value, $O_i$ is the observed value, and $\overline{Oi}$ is the observed value of the measured value. The model stability was measured according to the standard proposed by Jamieson [21]; that is: *n*-RMSE < 10%, model stability is excellent; 10% < *n*-RMSE < 20%, model stability is good; 20% < *n*-RMSE < 30%, model stability is general; *n*-RMSE > 30%, model stability is poor.

**Nitrogen nutrition index.**   The NNI was used to evaluate the nitrogen status of rice during its growth period in order to accurately reflect the nitrogen suitability in the rice plants.

$$\text{NNI} = \frac{N_a}{N_c}, \tag{4}$$

where $N_a$ is the measured value of the PNC, and $N_c$ is the critical value of nitrogen concentration. When NNI = 1, the state of nitrogen nutrition is the most appropriate, when NNI > 1, the nitrogen nutrition is in the state of excess, and on the contrary, when NNI < 1, the plant is in the state of under-nutrition.

## Data analysis

In this study, Excel 2010 software (Microsoft Corporation, Redmond, WA, USA) was used for data processing and calculation, IBM SPSS 22.0 software (SPSS, Inc., Chicago, IL, USA). was used for one-way ANOVA and multiple comparisons, and Origin 2018 software (OriginLab, Massachusetts, MA, USA) was used for illustrations. In the two experiments, the data from Experiment 1 were used to build the model, and the data from Experiment 2 were used to verify the model.

**Table 2. Dynamic changes in leaf dry matter and leaf nitrogen concentration of rice under different nitrogen application rates.**

| Cultivar | | Growth stage | LDM (t ha$^{-1}$) | | | | PNC (%) | | | |
|---|---|---|---|---|---|---|---|---|---|---|
| | | | Treatment | | | | | | | |
| | | | N0 | N75 | N150 | N225 | N0 | N75 | N150 | N225 |
| **Early rice** | 'Zhongjiazao 17' | TS | 0.49±0.03c | 0.75±0.32b | 0.79±0.08b | 0.81±0.01a | 3.20±0.13c | 3.46±0.04b | 3.78±0.06a | 3.76±0.10a |
| | | JS | 0.82±0.09d | 0.97±0.18c | 1.03±0.11b | 1.48±0.10a | 1.80±0.23d | 2.03±0.05c | 2.30±0.05b | 2.74±0.26a |
| | | BS | 1.18±0.16d | 1.58±0.11c | 1.85±0.23b | 1.95±0.02a | 1.55±0.34c | 1.42±0.09c | 1.86±0.31a | 1.77±0.32b |
| | | HS | 1.44±0.27d | 1.99±0.08c | 2.25±0.35b | 2.48±0.15a | 0.94±0.12c | 1.41±0.56a | 1.48±0.58a | 1.36±0.17b |
| | | FHS | 1.08±0.05d | 1.59±0.18c | 2.04±0.17b | 2.29±0.15a | 0.74±0.06d | 0.74±0.06c | 0.91±0.04b | 1.24±0.05a |
| | 'Changliangyou 173' | TS | 0.49±0.11d | 0.51±0.10c | 0.67±0.06b | 0.82±0.05a | 3.32±0.45b | 3.96±0.05a | 3.36±0.91b | 3.95±0.23a |
| | | JS | 0.79±0.15b | 1.02±0.10b | 1.26±0.03a | 1.31±0.04a | 1.75±0.06d | 1.88±0.29c | 2.53±0.09a | 2.33±0.07b |
| | | BS | 1.35±0.01d | 1.65±0.15c | 2.04±0.15a | 2.19±0.03b | 1.42±0.11d | 1.66±0.25c | 1.77±0.17b | 2.18±0.13a |
| | | HS | 1.62±0.04d | 1.92±0.28c | 2.44±0.32b | 2.55±0.14a | 1.09±0.18c | 1.36±0.11b | 1.63±0.24a | 1.60±0.37a |
| | | FHS | 1.31±0.13c | 1.53±0.20b | 2.33±0.34a | 2.43±0.08a | 0.68±0.07d | 0.83±0.04c | 1.04±0.08b | 1.36±0.31a |
| | | | N0 | N90 | N180 | N270 | N0 | N90 | N180 | N270 |
| **Late rice** | 'Fumeizhan' | TS | 1.12±0.12d | 1.33±0.28c | 1.61±0.15b | 1.93±0.09a | 2.41±0.10d | 2.56±0.06c | 2.65±0.07b | 3.06±0.48a |
| | | JS | 1.54±0.01d | 2.13±0.07c | 2.25±0.01b | 2.59±0.10a | 1.86±0.06c | 2.15±0.06b | 2.09±0.15a | 2.02±0.23a |
| | | BS | 2.20±0.26c | 2.75±0.12a | 2.82±0.09a | 3.38±0.33b | 1.14±0.11d | 1.58±0.09c | 1.74±0.03b | 1.87±0.02a |
| | | HS | 2.52±0.16b | 2.94±0.06b | 3.37±0.08b | 3.68±0.05a | 0.98±0.09d | 1.19±0.08c | 1.33±0.03b | 1.44±0.04a |
| | | FHS | 1.96±0.15d | 2.60±0.15c | 2.93±0.02b | 3.32±0.04a | 0.79±0.00d | 1.03±0.05c | 1.19±0.03b | 1.33±0.03a |
| | 'Taiyouhang 1573' | TS | 1.22±0.06d | 1.38±0.14c | 1.81±0.09b | 2.13±0.11a | 2.25±0.06c | 2.46±0.01b | 2.84±0.29a | 3.35±0.08a |
| | | JS | 1.99±0.15d | 2.24±0.19c | 2.28±0.05b | 2.73±0.22a | 1.40±0.06c | 1.72±0.07a | 1.69±0.00b | 1.94±0.22a |
| | | BS | 2.26±0.16c | 2.81±0.17b | 2.96±0.11a | 2.96±0.26a | 1.23±0.02c | 1.52±0.03b | 1.72±0.11b | 1.84±0.10a |
| | | HS | 2.98±0.07c | 3.20±0.09b | 3.39±0.04a | 3.39±0.04a | 1.07±0.06b | 1.25±0.01b | 1.42±0.03a | 1.56±0.03a |
| | | FHS | 1.95±0.07d | 2.44±0.06c | 3.02±0.16b | 3.29±0.08a | 0.80±0.04b | 0.96±0.04b | 1.16±0.09a | 1.38±0.06a |

In the table above, TS represents the tillering stage, JS represents the jointing stage, BS represents the booting stage, HS represents the heading stage, and FHS represents the full heading stage.

LDM, leaf dry matter; PNC, plant nitrogen concentration.

Data in the table represent the average value ± standard error, and those with the same letters are not significantly different (*P*<0.05).

# Results

## Dynamic changes in leaf dry matter mass and nitrogen concentration

The data in Table 2 show the dynamic accumulation process of LDM and rice PNC of different varieties of double-cropped rice under different nitrogen application levels. It can be seen from Table 2 that with rice growth and development, the dry matter weight of the leaves of the early and late rice increased continuously from the tillering stage and tended to stabilize until the highest value was reached at the heading stage, following which it decreased slightly at the full heading stage. However, the rice PNC decreased with the development of the growth period, and the changing trend of leaf dry matter and nitrogen concentration in early and late rice was similar. The LDM of 'Zhongjiazao 17' ranged from 0.46 to 2.64 t·ha$^{-1}$ and the PNC ranged from 0.66% to 3.85%, and the LDM of 'Changliangyou 173' ranged from 0.37 to 2.72 t ha$^{-1}$ and the PNC ranged from 0.61% to 4.17%. The LDM of 'Fumeizhan' ranged from 1.02 to 3.73 t ha$^{-1}$ and the PNC ranged from 0.78% to 3.62%, and the LDM of 'Taiyouhang 1573' ranged from 1.17 to 3.43 t ha$^{-1}$ and the PNC ranged from 0.76% to 3.43%.

## Critical nitrogen dilution curve based on leaf dry matter

According to the curve construction method proposed by Justes [20], the critical nitrogen concentration dilution curve of double-cropped rice was constructed based on the critical nitrogen content and the corresponding maximum leaf dry matter data points of the four double-cropped rice varieties in Experiment 1. The critical nitrogen concentration of the rice decreased with the increase in dry matter mass, and the changing trend could be fitted by the power function equation. Based on LDM, the critical nitrogen dilution curve models of the different varieties of early and late rice were constructed (Figs 1 and 2). In the early rice model of 2019, parameter *a* was 2.83 and 2.57 for 'Zhongjiazao 17' and 'Changliangyou 173', respectively, and parameter *b* was 0.83 and 0.69, respectively. In the late rice model, parameter *a* was 6.58 and 6.92 for 'Taiyouhang 1573' and 'Fumeizhan', respectively, and parameter *b* was 1.29 and 1.33, respectively.

The results of the 2020 data modeling in Experiment 1 were as follows (Fig 2): 'Zhongjiazao 17': $N_c\% = 2.47LDM^{-0.67}$, $R^2 = 0.83$; 'Changliangyou 173': $N_c\% = 2.62LDM^{-0.66}$, $R^2 = 0.82$; 'Taiyouhang 1573': $N_c\% = 2.38LDM^{-0.35}$, $R^2 = 0.92$; 'Fumeizhan': $N_c\% = 2.07LDM^{-0.39}$, $R^2 = 0.99$.

In order to further analyze the differences between the four early and late rice varieties, the power function models were linearized; that is, $ln(N_c) = ln\ a - b\ ln$ LDM. The variance analysis method was used to analyze the difference between the slope and intercept. The results showed that the *P*-value of the leaf slope and intercept between the two early rice varieties was greater than 0.05 ($P>0.05$), indicating that there was no significant difference. The *P*-value of the leaf slope and intercept of different varieties of late rice in 2019 was also not significantly different ($P>0.05$), but significantly different results were obtained in 2020 ($P = 0.012$). The preliminary analysis showed that during the period of late rice in 2020, especially from September to October, severe freezing and dew weather occurred in the experimental area, which led to the instability of the late rice model in this year. Therefore, the late rice model of this study was only based on the data from Experiment 1 in 2019. Totally, after synthesizing the critical N concentration dilution curves of the early and late rice, two fixed models of N concentration dilution curves for early and late rice were obtained:

Early rice: $N_c\% = 2.66LDM^{-0.79}$, $R^2 = 0.88$;

Late rice: $N_c\% = 7.46LDM^{-1.42}$, $R^2 = 0.91$ (data from Experiment 1 in 2019)

## Verification of the critical nitrogen dilution curve model

Independent data points were used to verify the model. In this study, the critical nitrogen concentration and the maximum leaf dry matter data points of the four double-cropped rice varieties in Experiment 2 were selected to verify the model. The maximum dry matter in the independent data group was substituted into the $N_c\%$ curve, and the simulated and measured values were compared. The fitting degree of the model was visually displayed using a 1:1 histogram. The simulated values of critical nitrogen content were calculated by substituting the measured dry matter data points into Eq (1), and the simulated values were compared with the observed values, as shown in Fig 3. Compared with the measured value and simulated value, the 1:1 graph was used to directly reflect the fitting degree of the model. The RMSE and *n*-RMSE values of the different varieties were calculated in Table 3. The RMSE values were all below 0.60 and the *n*-RMSE values were all less than 20%, which indicated that the established early and late rice models could accurately represent the nitrogen dilution process of the plant body and thus could be used to further evaluate the nitrogen nutrition status of double-cropped rice.

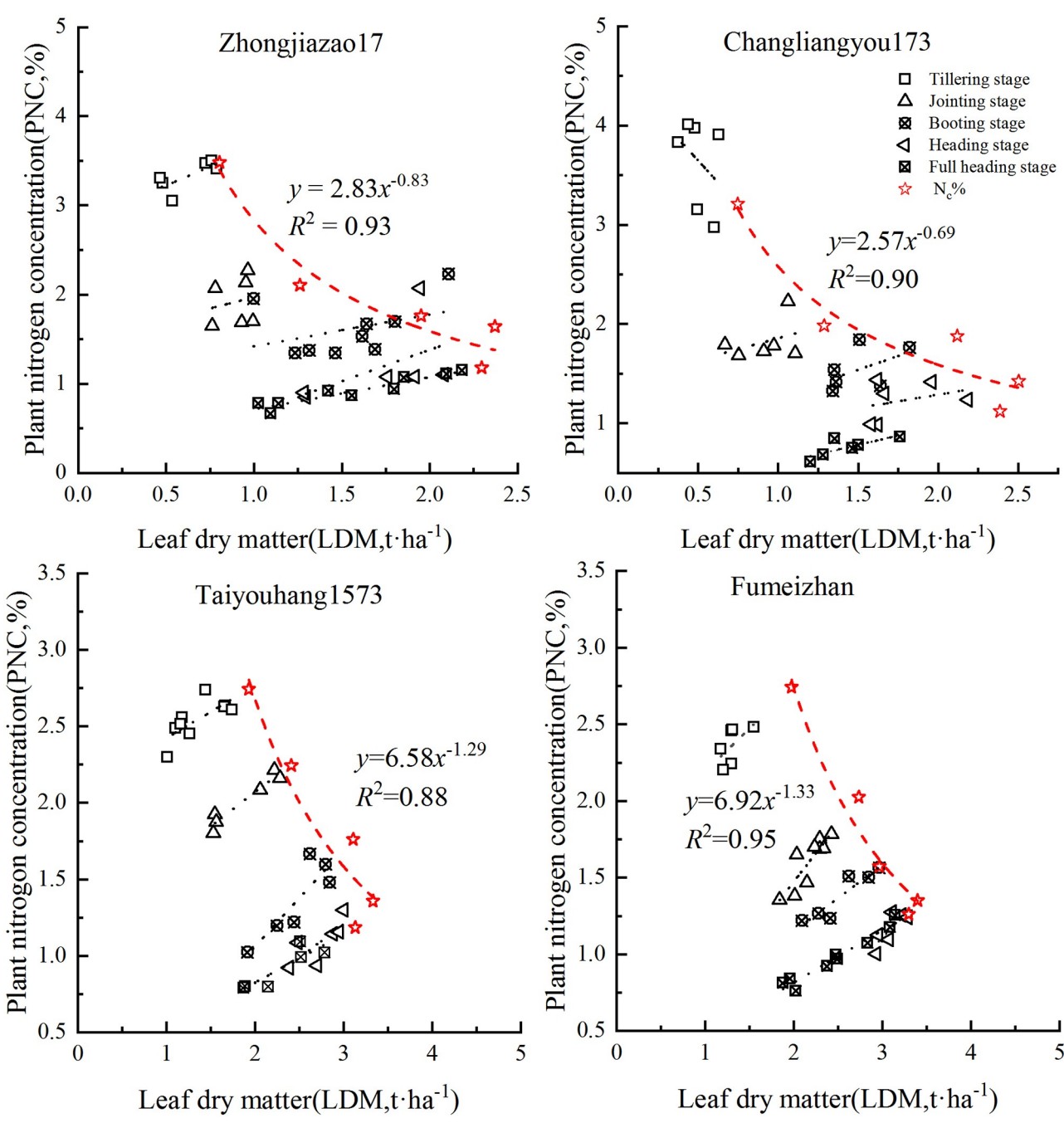

**Fig 1. Critical nitrogen concentration dilution curve of rice based on leaf dry matter in 2019.**

## Changes in the rice NNI during the growth stage

According to the calculation, the change range of the NNI of the two-year double-cropped rice under different nitrogen application rates was as follows (Fig 4). The NNI index increased with the increase of n application rate, and the NNI index of the same variety varied greatly with different growth periods and different years. While from the rice jointing stage to the booting stage, the NNI index all showed an increasing trend and then decreased to the heading

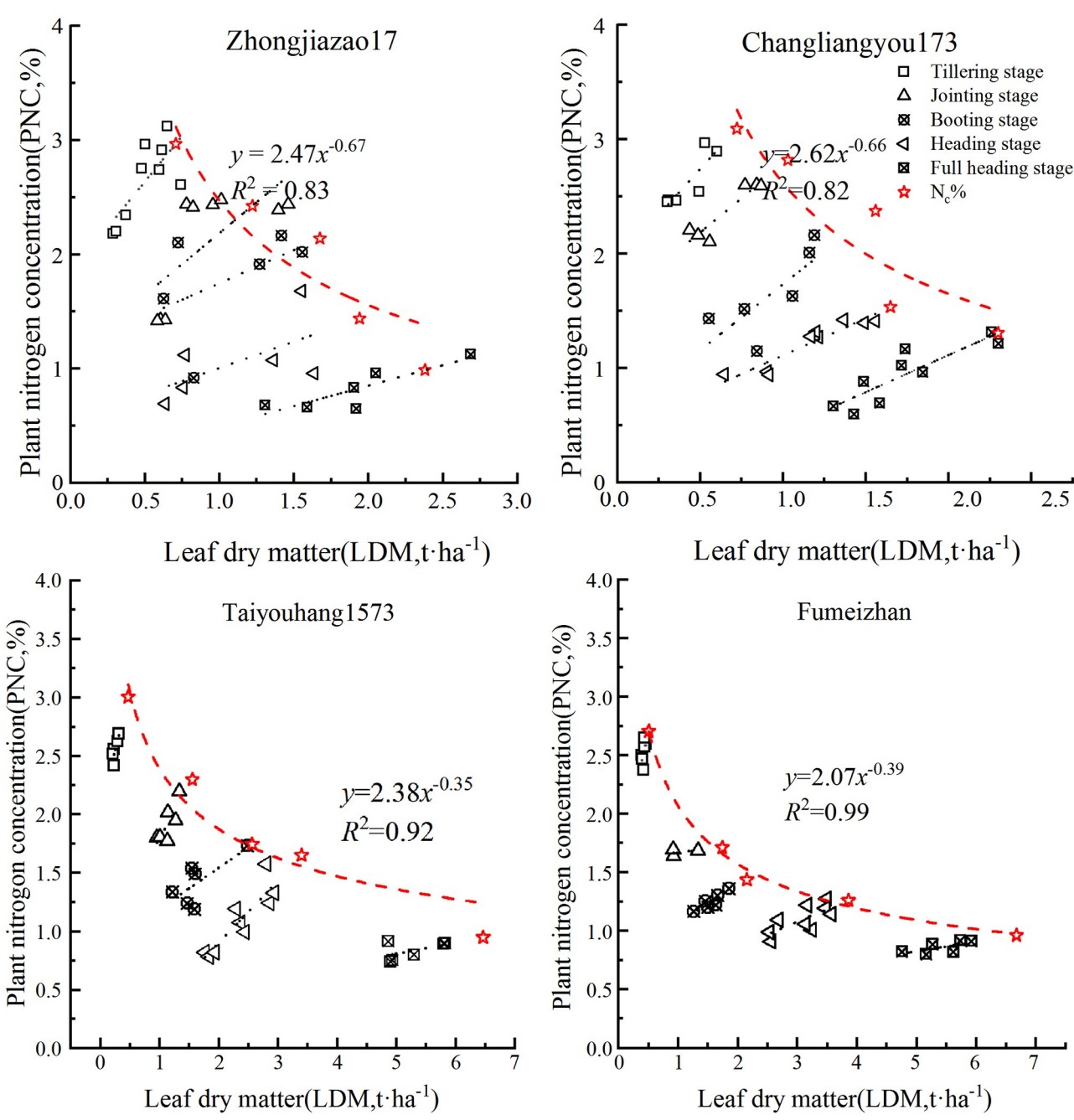

**Fig 2. Critical nitrogen concentration dilution curve of rice based on leaf dry matter in 2020.**

stage. As for early rice: N0: 0.54–0.69, N75: 0.73–1.05, N150: 1.02–1.38, N225: 1.17–1.63; late rice: N0: 0.27–0.58, N90: 0.50–0.89, N180: 0.74–1.15, and N270: 0.92–1.29. For early rice, according to the relationship between NNI and the value of 1, NNI of N0 treatment was significantly less than 1, NNI of N150 and N75 treatment were higher than or close to 1, NNI of N225 treatment was slightly higher than 1, which indicated that N150 and N75 treatment could meet the requirement of nitrogen for early rice growth. As for late rice, N0 and N90 were in deficit, N180 was moderate, and N270 was in excess.

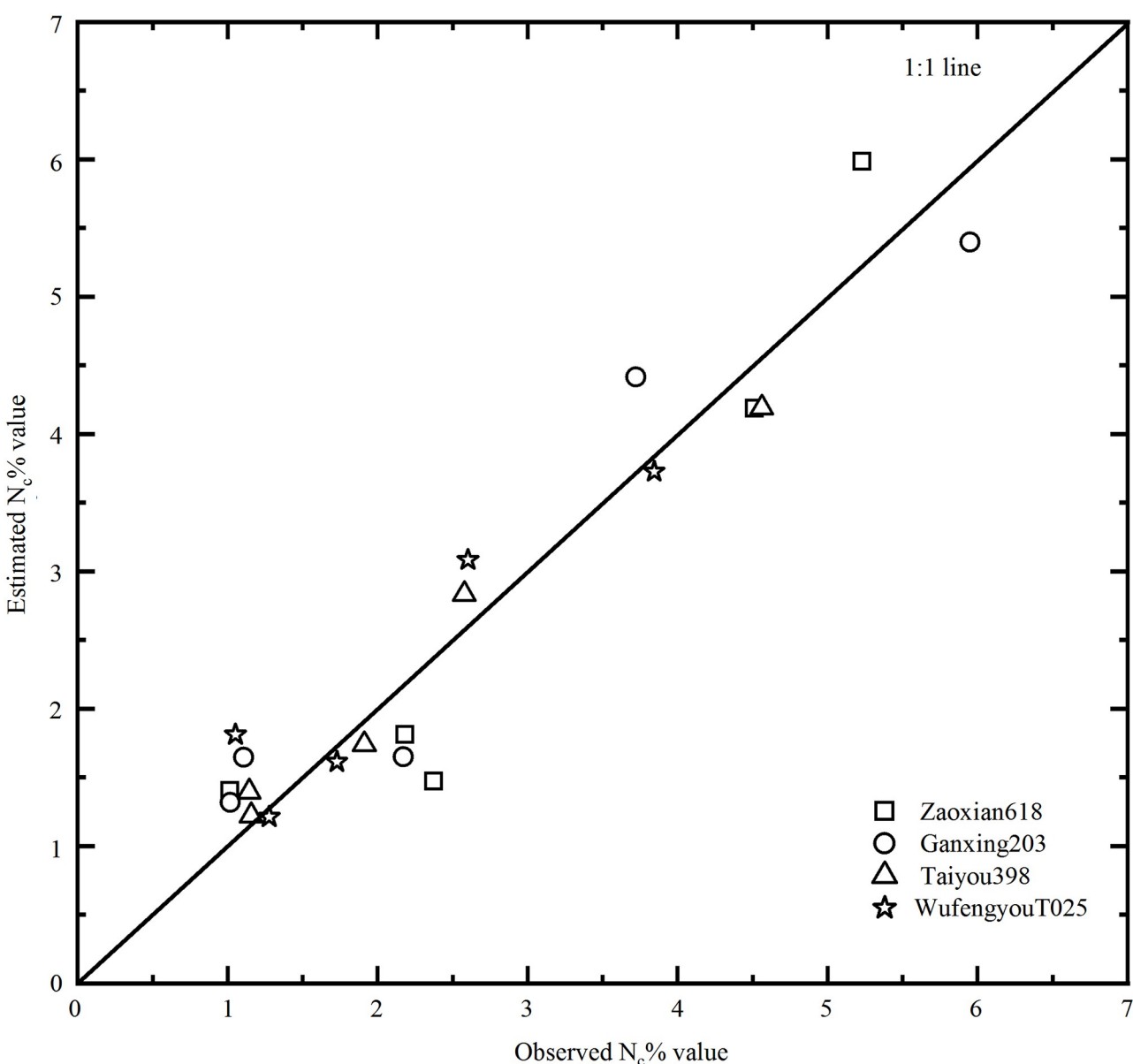

**Fig 3. Verification of critical nitrogen concentration dilution curve of double-cropped rice based on leaf dry matter (%).**

## Relationship between NNI and relative yield

The relationship between the NNI and relative yield (RY) of rice was studied. Fig 5 shows that the relationship between NNI and the RY of early and late rice was quadratic. The RY first

**Table 3. Parameter table of the model test results.**

| Indexes | Early rice | | Late rice | |
|---|---|---|---|---|
| | 'Zaoxian 618' | 'Ganxing 203' | 'Taiyou 398' | 'Wufengyou T025' |
| **RMSE** | 0.59 | 0.54 | 0.24 | 0.41 |
| ***n*-RMSE** | 19.45% | 19.25% | 10.78% | 19.49% |

RMSE, root mean square error; *n*-RMSE, relative root mean square error.

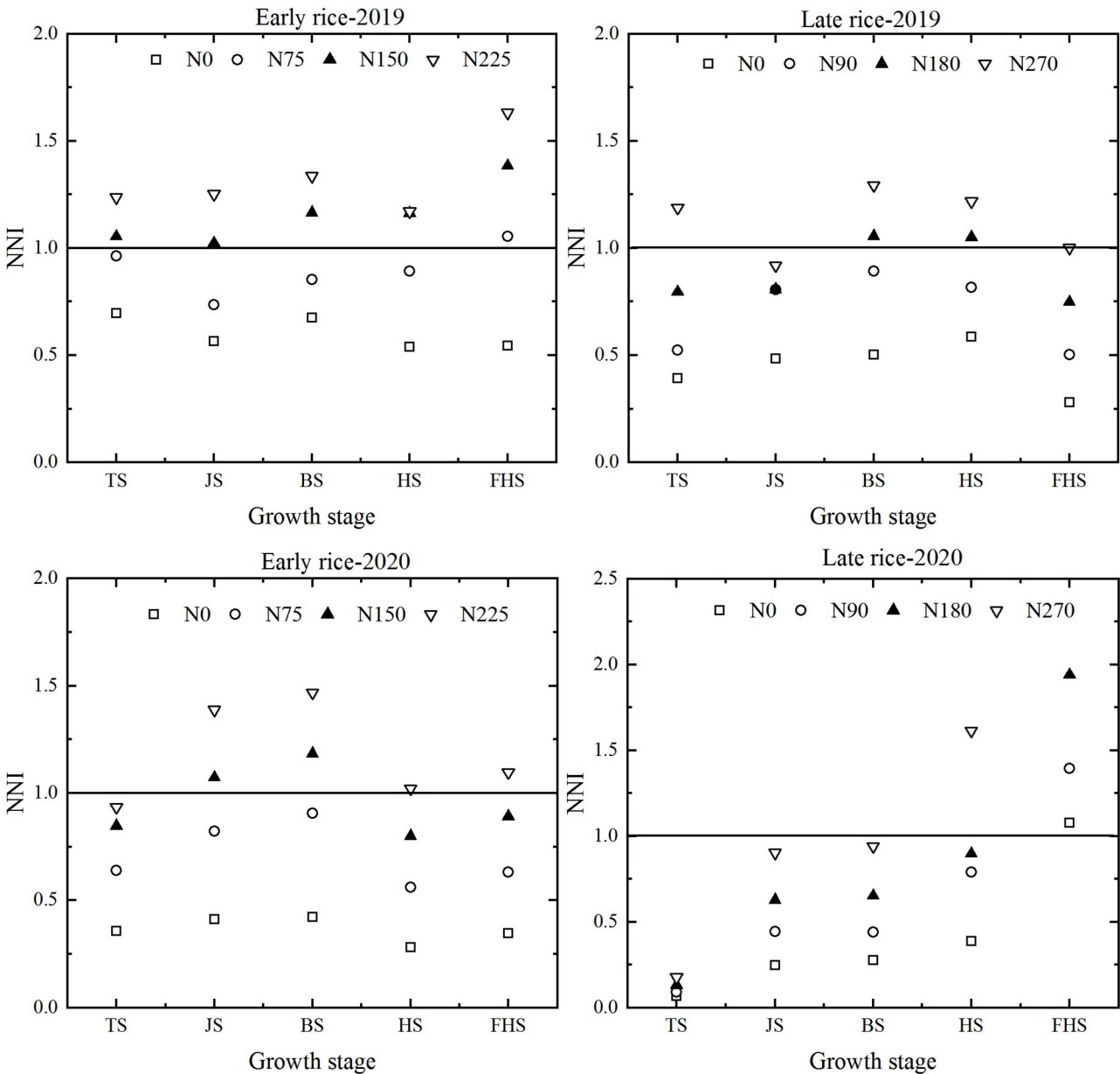

**Fig 4. Dynamic changes in the nitrogen nutrition index of rice under different nitrogen application rates.**

increased and then decreased with the increase in NNI. For early rice, the determination coefficient of the regression equation was highest at the heading stage, at 0.85, reaching a significant level. The maximum value of RY was 0.96 when NNI was 1.04. As for late rice, the coefficient of determination of the regression equation was highest at the booting stage, being 0.82, reaching a significant level. The maximum value of RY was 0.94 when those of NNI was 0.92.

## Discussion

The agronomic benefits of nitrogen application for increasing rice production have been recognized, and the critical nitrogen concentration dilution curve is an important crop nitrogen

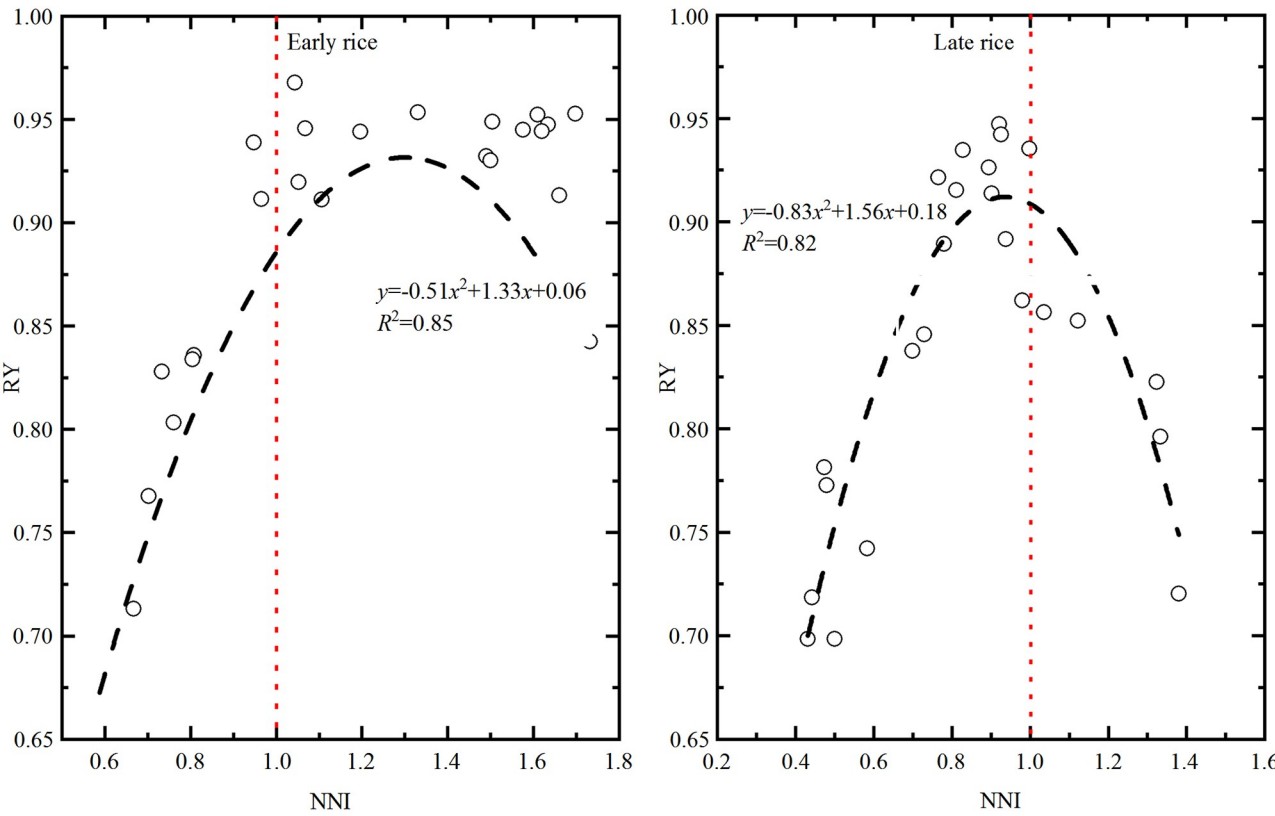

**Fig 5. Relationship between nitrogen nutrition index (NNI) and the Relative Yield (RY) of rice.**

nutrition diagnosis method that is used in many countries. Preliminary studies proposed that the nitrogen concentration dilution curve could be constructed by using the dry matter from different parts of crops [22–24]. In this study, the curve equation of critical nitrogen change in rice was established using the relationship between the leaf dry matter and nitrogen concentration of the LDM and PNC of the fractionated rice ($N_c$% = 2.66LDM$^{-0.79}$ and $N_c$% = 7.46LDM$^{-1.42}$ for early and late rice, respectively). In the critical nitrogen concentration dilution curve, parameter $a$ represents the plant nitrogen content when the dry matter is 1 t·ha$^{-1}$ and represents the intrinsic nitrogen demand characteristics in the early stage of crop growth [25]. Parameter $b$ represents the critical nitrogen concentration, which decreases with the increase in the dry matter mass in the upper part of the ground. In this study, the parameters $a$ and $b$ in the models of early rice were lower than those of the late rice models, which indicated that the nitrogen demand was large in the early stage of late rice growth. The reason may be that the temperature was higher in the late rice period, during which the rice grew fast and thus the process of nitrogen dilution was accelerated. At the same time, there were limited differences in the values of $a$ and $b$ of the critical nitrogen concentration dilution curves in different years, among which the difference in the models in early rice was not significant, while the values of $a$ and $b$ in the model of late rice 2020 were smaller than in 2019. The reason for the great difference may be that there was colder and more exposed windy weather in Jiangxi Province from late September to early October in 2020, which resulted in the decrease in the nitrogen dilution rate of the rice leaves, causing the parameters $a$ and $b$ to be significantly lower than that of the previous year. This result is consistent with that of Shimono et al. [26].

**Table 4. The model comparison results.**

| Region | Parameter $a$ | Parameter $b$ | Model accuracy | Reference |
|---|---|---|---|---|
| Northern rice growing area in China | 1.96 | 0.56 | 0.87 | Song et al. [28] |
| Double-cropped rice area in China | Early rice: 3.37 | 0.44 | 0.82 | He et al. [29] |
| | Late rice: 3.69 | 0.34 | 0.78 | |
| Rice wheat rotation area in Central China | 3.33 | 0.26 | 0.86 | Wang et al. [17] |

Different rice genotypes have different sensitivities to nitrogen fertilizer [27]. In this study, parameter $b$ from rice variety 'Zhongjiazao' was greater than that from rice variety 'Changliangyou 173', parameter $b$ from rice variety 'Fumeizhan' was smaller than that from rice variety 'Taiyouhang 1573', while parameter $a$ exhibited uncertainty. Compared with previous studies (Table 4), the $a$ value of rice in south China was higher than that in north China, may be because low temperature led to low nitrogen uptake, so nitrogen accumulation in north China was lower [28]. The parameter $a$ in this study was lower than that in reference [17, 29], and according to Lemaire [9], the correlation between the initial N uptake capacity of crops and the amount of N supplied by soil at the early growth stage can also be considered as a reason for the difference. The value $b$ from early and late rice was all greater than those of the literature, and the difference between the curve and the curve constructed in the northern and central rice areas was obvious [17, 29]. The reason may be that the leaf dry matter weight is less than the plant dry matter weight, which leads to the significant increase of the ratio of plant nitrogen nutrition to dry matter. These above results indicated that the regional impact model parameters need to be established in different regions.

NNI is widely used in crop nitrogen nutrition diagnosis, but it is difficult to achieve NNI = 1 in agricultural production. Plant sensitivity to nitrogen fertilizer differs across seasons. The results from this research show that the optimum nitrogen application rate of early rice was slightly smaller than 150 kg·ha$^{-1}$ and of late rice was slightly less than 180 kg·ha$^{-1}$, respectively. As for this study, the optimum nitrogen application rate of rice was similar to that of local experience fertilization (165–180 kg·ha$^{-1}$), and was similar to that of Zhu et al. [30]. who proposed the total nitrogen application (105 kg·ha$^{-1}$ for early rice and 146 kg·ha$^{-1}$ for late rice) could achieve the goal of high yield. While only four nitrogen application rates were set in both early and late rice in this study, so the optimal nitrogen application rate was much less explored, which needs further study. According to the relationship between NNI and relative yield, the relative yield was 95% of the maximum yield when NNI was within the suitable range, which indicated that proper fertilization in double cropping rice of Jiangxi Province was beneficial to the increase of rice yield.

## Conclusion

The plant nitrogen content of double-cropped rice in this study decreased with the increase in LDM, and different rice genotypes and plant seasons had different sensitivities to nitrogen fertilizer. In this study, the critical nitrogen dilution curve of double-cropped rice in Jiangxi based on LDM was established using four varieties across two years, which confirmed the stability of the parameters. The model can be used to diagnose the nitrogen nutrition status of Jiangxi rice region and effectively guide the scientific application of nitrogen fertilizer. According to the NNI model established in this study, it is recommended that the optimal nitrogen application rate for early rice is sightly smaller than 150 kg·ha$^{-1}$ and that for late rice is about 180 kg·ha$^{-1}$ in the study area. The nitrogen application rate of farmers in this area is high, and

thus the potential for reducing fertilizer application based on these findings in this study is great.

## Supporting information

**S1 Data.**
(XLS)

## Acknowledgments

We thank LetPub (www.letpub.com) for its linguistic assistance during the preparation of this manuscript.

## Author Contributions

**Conceptualization:** Ying Liu.

**Data curation:** Jizhong Liu, Shifu Shu.

**Funding acquisition:** Jizhong Liu, Yanda Li.

**Methodology:** Binfeng Sun.

**Resources:** Yanda Li.

**Supervision:** Luofa Wu.

**Writing – original draft:** Chun Ye.

**Writing – review & editing:** Chun Ye.

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
