## [Decision Letter · Decision Letter 0]

10 Jun 2021

PONE-D-21-10238

Simulation of the critical nitrogen dilution curve in double-cropped rice based on leaf dry matter weight

PLOS ONE

Dear Dr. Chun,

Thank you for submitting your manuscript to PLOS ONE. After careful consideration, we feel that it has merit but does not fully meet PLOS ONE’s publication criteria as it currently stands. Therefore, we invite you to submit a revised version of the manuscript that addresses the points raised during the review process.

We look forward to receiving your revised manuscript.

Kind regards,

Dali Zeng

Academic Editor

PLOS ONE

Journal Requirements:

Additional Editor Comments (if provided):

We encourage authors to add more rice varieties for this research, or repeat the research in another year or place before submitting again.

Reviewers' comments:

Reviewer's Responses to Questions

**Comments to the Author**

1. Is the manuscript technically sound, and do the data support the conclusions?

Reviewer #1: Partly

Reviewer #2: Yes

2. Has the statistical analysis been performed appropriately and rigorously? 

Reviewer #1: No

Reviewer #2: Yes

3. Have the authors made all data underlying the findings in their manuscript fully available?

Reviewer #1: No

Reviewer #2: Yes

4. Is the manuscript presented in an intelligible fashion and written in standard English?

Reviewer #1: Yes

Reviewer #2: Yes

5. Review Comments to the Author

Reviewer #1: Rice is the staple food of more than half the world's population. Improving nitrogen use efficiency to increase rice yield has played an important role in promoting sustainable agricultural development. Increasing nitrogen application to increase rice yield has been widely used in modern agriculture. The relationship between nitrogen use and rice yield has always been the focus of attention of rice cultivators and breeders. Based on the accumulation of dry matter in the leaves, the study on the most suitable nitrogen application amount of double cropping rice is beneficial to the protection of environment and the sustainable development of agriculture. However, there are still some deficiencies that need to be further refined before publication.

1. The biggest defect of this manuscript is that the author only used two early rice varieties and two late rice varieties to carry out the experiment. Is the sample representative enough? Is it reasonable to use the data obtained from different nitrogen gradients of two varieties for statistical analysis and model establishment?

2. According to the above problems, the scope of the topics in the manuscript is too large and needs to be adjusted.

3. The ultimate goal of using nitrogen fertilizer is to improve yield. This paper analyzes the relationship between leaf dry matter weight and leaf nitrogen content, and the relationship between NNI and yield, so as to evaluate the best nitrogen application rate for high yield. However, due to the significant difference of nutrient transport efficiency among different varieties, it is suggested that the relationship between leaf dry matter weight, total N and yield should be analyzed.

4. It is shown in Figure 4 that there are great differences in NNI among different varieties or in different growth stages of the same variety. How does the author get the conclusion that "nitrogen absorption in the double cropped rice mainly occurs in the vegetable growth stage (joining stage and booting stage), and the nitrogen nutrition status at this stage directly determines the final yield and composition"? It needs to be supported by reasonable data or references.

5. Improve the quality of the text, such as "nitrogen nutrition index" and "NNI" in many parts of the text, which is suggested to be unified; in Table 1, the name of variety is wrongly written, such as "changiangyou 173", …… etc.

6. The definition of the picture in the paper is poor, so it is suggested to replace the picture with higher definition.

Reviewer #2: Review on the publication by Chun et al. under the title 'Simulation of the critical nitrogen dilution curve in double-cropped rice based on leaf dry matter weight.'

The article is connected with agronomically important species. The article is perfectly prepared. It's contain a lot of new and interesting results for scientific community. I do recommend it for publication

6. PLOS authors have the option to publish the peer review history of their article (what does this mean?). If published, this will include your full peer review and any attached files.

Reviewer #1: No

Reviewer #2: No

---

## [Author Response · Author response to Decision Letter 0]

22 Jul 2021

Dear Editors and Reviewers,

Thank you for your letter and for the reviewer’s comments concerning our manuscript entitled " Simulation of the critical nitrogen dilution curve in double-cropped rice based on leaf dry matter weight" (ID: PONE-D-21-10238). Those comments are all valuable and helpful for revising and improving our paper, as well as important guiding significance to our research. We have studied comments carefully and have made correction which we hope meet with approval. Revised portion are marked in red in the paper and the reviewers’ comments are responded as follows:

---

## [Decision Letter · Decision Letter 1]

15 Oct 2021

Simulation of the critical nitrogen dilution curve in Jiangxi double-cropped rice region based on leaf dry matter weight

PONE-D-21-10238R1

Dear Dr. Chun,

We’re pleased to inform you that your manuscript has been judged scientifically suitable for publication and will be formally accepted for publication once it meets all outstanding technical requirements.

Kind regards,

Dali Zeng

Academic Editor

PLOS ONE

Additional Editor Comments (optional):

Reviewers' comments:

Reviewer's Responses to Questions

**Comments to the Author**

1. If the authors have adequately addressed your comments raised in a previous round of review and you feel that this manuscript is now acceptable for publication, you may indicate that here to bypass the “Comments to the Author” section, enter your conflict of interest statement in the “Confidential to Editor” section, and submit your "Accept" recommendation.

Reviewer #1: (No Response)

Reviewer #2: All comments have been addressed

Reviewer #3: All comments have been addressed

2. Is the manuscript technically sound, and do the data support the conclusions?

Reviewer #1: No

Reviewer #2: Yes

Reviewer #3: Yes

3. Has the statistical analysis been performed appropriately and rigorously? 

Reviewer #1: Yes

Reviewer #2: Yes

Reviewer #3: Yes

4. Have the authors made all data underlying the findings in their manuscript fully available?

Reviewer #1: Yes

Reviewer #2: Yes

Reviewer #3: Yes

5. Is the manuscript presented in an intelligible fashion and written in standard English?

Reviewer #1: Yes

Reviewer #2: Yes

Reviewer #3: Yes

6. Review Comments to the Author

Reviewer #1: The local main rice varieties were specific and could not represent the diversity of early and late rice varieties.The small sample size is still the fatal shortcoming of this paper.

Reviewer #2: All my previous concerns have been well addressed in this revised version. This work reported the developed critical nitrogen concentration dilution curves, based on leaf dry matter, were able to diagnose nitrogen nutrition in the double-cropped rice region, and the results are interesting and meaningful.

Reviewer #3: (No Response)

7. PLOS authors have the option to publish the peer review history of their article (what does this mean?). If published, this will include your full peer review and any attached files.

Reviewer #1: No

Reviewer #2: No

Reviewer #3: No

---

## [Editor Report · Acceptance letter]

22 Oct 2021

PONE-D-21-10238R1 

Simulation of the critical nitrogen dilution curve in Jiangxi double-cropped rice region based on leaf dry matter weight 

Dear Dr. Ye:

I'm pleased to inform you that your manuscript has been deemed suitable for publication in PLOS ONE. Congratulations! Your manuscript is now with our production department. 

Kind regards, 

on behalf of

Dr. Dali Zeng 

Academic Editor

PLOS ONE